# Peptidoglycan sensing by octopaminergic neurons modulates *Drosophila* oviposition

**C Leopold Kurz†, Bernard Charroux†, Delphine Chaduli, Annelise Viallat-Lieutaud, Julien Royet***

Aix-Marseille Université, Centre National de la Recherche Scientifique, UMR 7288, Institut de Biologie du Développement de Marseille, Marseille Cedex, France

**Abstract** As infectious diseases pose a threat to host integrity, eukaryotes have evolved mechanisms to eliminate pathogens. In addition to develop strategies reducing infection, animals can engage in behaviors that lower the impact of the infection. The molecular mechanisms by which microbes impact host behavior are not well understood. We demonstrate that bacterial infection of *Drosophila* females reduces oviposition and that peptidoglycan, the component that activates *Drosophila* antibacterial response, is also the elicitor of this behavioral change. We show that peptidoglycan regulates egg-laying rate by activating NF-κB signaling pathway in octopaminergic neurons and that, a dedicated peptidoglycan degrading enzyme acts in these neurons to buffer this behavioral response. This study shows that a unique ligand and signaling cascade are used in immune cells to mount an immune response and in neurons to control fly behavior following infection. This may represent a case of behavioral immunity.

*For correspondence: julien.royet@univ-amu.fr

†These authors contributed equally to this work

**Competing interests:** The authors declare that no competing interests exist.

## Introduction

As infectious diseases pose a threat to the host integrity and to its reproductive fitness, eukaryotes have evolved sophisticated mechanisms that detect and mobilize defenses against pathogens. Innate immunity, which is common to all metazoans, serves as a first-line defense against microbes. Its hallmarks are the recognition of microorganisms by germline-encoded non-rearranging receptors followed by rapid effector mechanisms such as phagocytosis, proteolytic cascade activation and production of antimicrobial peptides (*Hoffmann et al., 1999*). Genetic screens have identified two NF-κB-dependent pathways, IMD and Toll, as the main signaling cascades that control immune gene induction in *Drosophila* (*Buchon et al., 2014*; *Lemaitre et al., 1995*, *1996*; *Ferrandon et al., 2007*). Both pathways are activated by a unique ligand, the bacterial cell wall peptidoglycan (PGN). PGN from Gram-negative bacteria and bacilli preferentially activates the IMD pathway via its detection by two members of the same PeptidoGlycan Recognition Protein (PGRP) family, PGRP-LC and PGRP-LE. While PGRP-LC functions as the transmembrane signaling receptor upstream of the IMD pathway, PGRP-LE is detecting PGN intracellularly (*Bosco-Drayon et al., 2012*; *Choe et al., 2005*; *Gottar et al., 2002*; *Rämet et al., 2002*; *Kaneko et al., 2006*). PGN from Gram-positive bacteria rather engages a Toll-dependent response involving a circulating PGRP as sensor, PGRP-SA (*Michel et al., 2001*; *Leulier et al., 2003*). Indeed, in contrast to mammalian Toll-Like-Receptors, *Drosophila* Toll has no capacity to directly bind microbial motifs (*Brough et al., 2008*).

In addition to activate direct antimicrobial strategies, eukaryotes have developed behavioral mechanisms that facilitate the avoidance of pathogens or lower the impact of the infection. These phenotypes grouped under the term 'behavioral immunity' (*de Roode and Lefèvre, 2012*) or 'sickness behavior' (*Hart, 1988*) refer to a suite of neuronal mechanisms that allow organisms to detect the potential presence of disease-causing agents and to engage in behaviors which prevent contact with the invaders or reduce the consequences of the infection. Although such microbe-induced

**eLife digest** Bacteria are all around us: they are on our skin, in the food that we eat and inside our bodies, particularly in the gut. While many of these bacteria are harmless and some even help us digest our food, others can make us ill. Upon detecting harmful bacteria, our bodies therefore trigger an immune response intended to destroy them.

Some insects – including butterflies, moths and grasshoppers – have an additional way of defending themselves against bacteria besides their immune response. Whenever they detect harmful microorganisms, the insects change their behavior so as to reduce their chances of becoming infected and limit the damage an infection would cause. The insects move away from areas containing harmful bacteria, for example, and temporarily stop eating. But whereas the insects' immune response to bacteria is well documented, little was known about the mechanisms that underlie these changes in behavior.

Kurz, Charroux et al. set out to rectify this using another insect species, the fruit fly *Drosophila*. Flies that are infected with bacteria lay fewer eggs than healthy flies: a change in behavior that helps protect the offspring from infection. Kurz, Charroux et al. show that fruit flies are able to detect a component of the cell wall that surrounds all bacteria. This substance, known as peptidoglycan, activates a set of neurons in the fly that produce a chemical called octopamine. These neurons in turn activate a signaling pathway featuring a molecule known as NF-κB, and this causes the flies to lay fewer eggs.

Notably, peptidoglycan and NF-κB are also the molecules that trigger the anti-bacterial immune response. Fruit flies thus use the same pathway in immune cells and in neurons to trigger immune responses and behavioral changes, respectively. The challenge now is to identify precisely which neurons respond to bacterial peptidoglycan, and to work out how peptidoglycan changes the activity of these cells. Furthermore, studies have recently shown that bacterial peptidoglycan can influence the development of the mouse brain, as well as mouse behavior. This suggests that mechanisms for detecting harmful bacteria may be conserved across evolution, a possibility that requires further investigation.

behavioral changes have been reported in Lepidoptera and Orthoptera, deciphering the molecular mechanisms involved is experimentally challenging in these insects (*Sullivan et al., 2016*; *Kazlauskas et al., 2016*; *Adamo et al., 2007*; *Adamo, 2005*). Indeed, such an analysis requires a model organism with genetic tools allowing the manipulation of actors and regulators of both the immune and neuronal systems. Recent reports, mainly in *Drosophila*, start to unravel some aspects of these peculiar host-microbe interactions. Stensmyr et al. demonstrated that *Drosophila* avoid food contaminated by pathogenic bacteria by using an olfactory pathway exquisitely tuned to a single microbial odor, Geosmin (*Stensmyr et al., 2012*). Produced by harmful microorganisms, Geosmin is detected by specific *Drosophila* olfactory sensory neurons which then transfer the message to higher brain centers. Activation of this olfactory circuit ultimately induces an avoidance response, and suppresses egg-laying and feeding behaviors, thereby reducing the infection risk of both the adult flies and their offspring. *Drosophila* not only modify their behavior to avoid contamination by microbes or parasites, but also once they have been contaminated in order to reduce the impact of infection. For instance, direct exposure to bacteria impacts sleep patterns and induces hygienic grooming (*Kuo et al., 2010*; *Yanagawa et al., 2014*). In addition, *Drosophila* plastically increases the production of recombinant offspring in response to parasite infection (*Singh et al., 2015*). Although certainly involving a neuro-immunological integration, these microbe-induced behavioral changes are rarely understood at the molecular level, namely with no information on the nature of the elicitor and on the cellular and molecular machineries that link bacteria detection to behavioral changes. Moreover, canonical immune signaling pathways were never reported as being involved in those processes.

In order to appreciate further the interactions between infecting microbes and the host nervous system, we studied the effects of bacterial infection on one specific *Drosophila* behavior, female oviposition. We show here that fly exposure to bacteria is sufficient to trigger an egg-laying reduction

and identify bacteria cell wall PGN as the elicitor of this effect. We demonstrate that PGN-dependent NF−κB pathway activation in octopaminergic neurons is the molecular signal that triggers oviposition drop post-infection. Finally, we present data showing that this NF-κB pathway activity in neurons is fine-tuned by a specific isoform of the PGN-degrading enzyme PGRP-LB, thereby modulating a process that would otherwise leads to an exacerbated and detrimental egg-laying decrease. Altogether, this study shows that detection of a unique microbe-associated pattern is not only protecting the infected individual via immune response activation but is also reducing the impact of the bacteria on the whole population by decreasing oviposition on contaminated environments. This could constitute a case of behavioral immunity.

## Results

### Bacteria infection decreases fly oviposition

In *Drosophila,* infection by extracellular bacteria triggers a battery of cellular and humoral immune responses among which the IMD/NF-κB signaling cascade is of paramount importance. Direct recognition of bacterial peptidoglycan by membrane associated PGRP-LC or intracytoplasmic PGRP-LE triggers NF-κB nuclear translocation and signaling (*Figure 1A*). By degrading PGN, amidases such as PGRP-LB prevent over-activation of the IMD pathway and adapt the response to the dynamic of infection (*Figure 1A*) (*Zaidman-Rémy et al., 2006*). To test whether bacteria infection has physiological consequences other than strictly antimicrobial, we analyzed the egg-laying behavior of infected flies. Septic injury with a needle contaminated with the innocuous Gram-negative bacteria *E. coli*, induced a strong egg-laying drop within 6 hr following infection (*Figure 1B and C* and *Figure 1—figure supplement 1A*). The difference in the number of eggs laid by infected versus non-infected wild-type females was no longer seen after 24 hr demonstrating that the effects are transient and reversible (*Figure 1C* and *Figure 1—figure supplement 1A*). As for the immune response, bacteria-induced egg-laying drop was stronger and lasted longer in *PGRP-LB* mutants than in wild-type flies while the amount of bacteria within the animals were equivalent (*Figure 1C* and *Figure 1—figure supplements 1B*, *2A and B*) (*Zaidman-Rémy et al., 2006*). This effect was dose-dependent and not observed after sterile wound or when flies were injected with water (*Figure 1D and E*). This demonstrates that the effects were due to the presence of the bacteria in the body cavity and not to cuticle wounding (*Figure 1C* and *Figure 1—figure supplement 1*). Consequently, we thereafter used untreated animals as uninfected controls. Egg-laying reduction was also observed when flies were orally infected with bacteria and when flies mutant for another IMD pathway antagonist, *Pirk*, were infected (*Figure 1—figure supplement 2C and D*). Altogether, these results demonstrate that bacteria infection induces an egg-laying decrease in wild-type flies that is exacerbated in *PGRP-LB* mutants.

### PGN blocks egg laying by activating NF-κB pathway

Knowing the role of PGRP-LB as a scavenging receptor for bacterial PGN and of PGN as an IMD pathway upstream ligand, we hypothesized that PGN could be the bacterial elicitor triggering egg-laying drop. Importantly, injection of highly purified *E. coli* PGN in both wild type and *PGRP-LB* mutants fully phenocopied the effects seen with injection of bacteria (*Figure 1D and E*). To further confirm the implication of the IMD pathway, we tested the effect of bacteria infection on egg-laying behavior of IMD pathway mutants. Mutants for the transcription factor Relish (*Rel^{E20}*) and for the caspase Dredd (*Dredd^{D55}*) did not present egg-laying drop 6 hr post-infection (*Figure 1F*). In addition, we performed epistasic analyses with *PGRP-LB* mutants and demonstrated that inactivation of, *Relish*, *Dredd* or the intracytoplasmic receptor *PGRP-LE*, but not the transmembrane protein *PGRP-LC*, completely suppressed the *PGRP-LB*-dependent egg-laying drop (*Figure 1G*). Altogether these results demonstrate that PGRP-LE-dependent PGN detection by the fly induces a reduction of female oviposition by triggering NF-κB pathway activation.

### Bacteria infection induces a temporary retention of oocytes within ovaries

Previous studies reported that flies infected with pathogenic bacteria (*Salmonella typhimurium*, *Burkholderia cenocepacia* and *Serratia marcescens*) present an oviposition defect consecutive to ovary

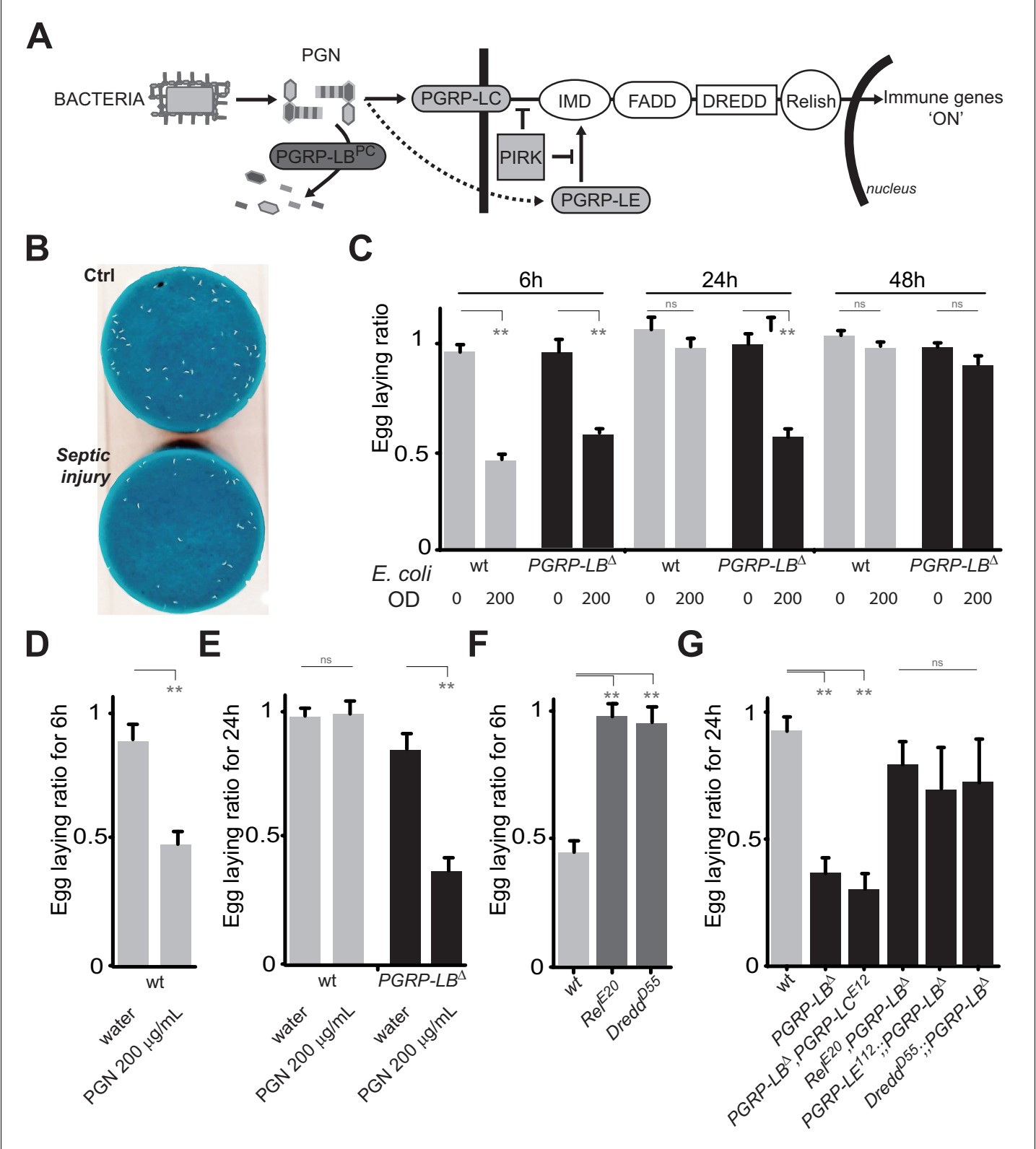

**Figure 1.** PGN-mediated NF-κB pathway activation decreases female oviposition. (**A**) Schematic representation of the *Drosophila* IMD pathway. Peptidoglycan (PGN) released by bacteria is recognized at the cell membrane by PGRP-LC or inside the cell by PGRP-LE. This PGRP/PGN interaction triggers via IMD, FADD, DREDD, nuclear translocation of Relish. Extracellular PGRP with amidase activity, such as PGRP-LB[PC], dampens this signaling by degrading PGN. (**B**) Septic injury impairs egg-laying capacity in wild-type females. Pictures of fly tubes seen from the top. The blue dye is used to

*Figure 1 continued*

facilitate quantification of the eggs that appear as white dots. (**C**) Septic injury transiently impairs egg-laying capacity in wild type and in *PGRP-LB* mutant females. The egg-laying ratio for a time window (6 hr or 24 hr) corresponds to the number of eggs laid by a female after infection over the number of eggs laid by control female of the same genotype. (**D and E**) Injection of highly purified PGN in wt (**D, E**) and *PGRP-LB* mutants (**E**) is sufficient to reduce egg laying. (**F**) Mutation in Relish or in Dredd is preventing egg laying decrease post-infection. (**G**) Epistatic analyses showing that mutations in PGRP-LE, NF-κB or Dredd, but not in PGRP-LC are rescuing *PGRP-LB* mutant phenotype. For **C, D, E, F** and **G**; shown is the average egg-laying ratio ± SEM from at least two independent trials with at least 20 females per genotype and condition used. * indicates p<0.01; ** indicates p<0.001; n.s. indicates p>0.05, unpaired two-tailed Mann-Whitney test versus indicated controls.

The following figure supplements are available for figure 1:

**Figure supplement 1.** Infection by bacteria decreases female oviposition.

**Figure supplement 2.** IMD pathway overactivation leads to an exacerbated oviposition drop following exposure to bacteria.

**Figure supplement 3.** Bacterial infection provokes an accumulation of mature oocytes in female ovaries.

degeneration (*Brandt and Schneider, 2007*). To identify the causes of the bacteria-induced egg-laying reduction, we dissected ovaries from untreated flies, flies stabbed only or injured with a needle contaminated with bacteria, and counted the numbers of oocytes of different maturation stages. Six hours post-exposure to *E. coli*, ovaries from wild-type flies contained three-times more mature stage 14 oocytes than uninfected animals and less stage 10 immature oocytes (*Figure 1—figure supplement 3A,B and C*). Egg-laying drop was reversible by 24 hr suggesting that the exposure to bacteria does not prevent oocyte maturation in the ovaries but rather blocks egg release into the oviducts, leading to an accumulation of mature oocytes in ovaries of infected flies. Since an oviposition drop could be the result of a stage-specific oocyte apoptotic death, we tested this hypothesis. Antibody staining against activated-caspase (DCP1) did not reveal any sign of increased apoptosis in ovaries from infected females compared to controls (*Figure 1—figure supplement 3C and D*). These results demonstrate that the presence of bacteria within the female body cavity induces a transient retention of mature oocytes within the ovaries leading to a drop of egg laying. We then tested whether the ovulation defect was not an indirect consequence of an infection-dependent reduced food intake. As shown *Figure 1—figure supplement 3E*, bacterial infection did not impair adult feeding behavior in both wild-type and *Relish* mutants 6 hr after bacterial pricking.

## Immune and behavioral responses to PGN are controlled by distinct PGRP-LB isoforms

To identify the tissue(s) in which IMD pathway activation modulate egg-laying behavior in response to infection, we looked for the site(s) of action of the PGRP-LB amidase. Indeed, as an exacerbated egg-laying drop post-infection might be detrimental, we hypothesized that the cells in which PGRP-LB is modulating the immune pathway could be the ones in which IMD activation is interfering with egg-laying. Interestingly, the PGRP-LB locus generates three protein isoforms (PGRP-LB [PA, PC, PD]) that all have in common the enzymatic amidase domain (*Figure 2A*). While the secreted PGRP-LB[PC] cleaves the PGN in the extracellular space (*Paredes et al., 2011*; *Zaidman-Rémy et al., 2006*), the site and mode of action of the two other isoforms are unknown. In contrast to PGRP-LB[PC], neither PGRP-LB[PA] nor PGRP-LB[PD] possess a signal peptide, suggesting that they could act intracellularly. PGRP-LB[PA] is made of the amidase domain only and PGRP-LB[PD] contains an additional short N-terminal peptide of unknown function (*Figure 2A*). To test the implication of these isoforms in immune and behavioral responses, we tested their ability to rescue *PGRP-LB* null mutant phenotypes. For that purpose, we cloned 3 Kb, 2.8 Kb and 3.8 Kb of genomic DNA upstream of the predicted transcription start site of each isoform upstream of the Gal4 coding sequences (*Figure 2A*). These lines (called pLB1[Gal4], pLB2[Gal4] and pLB3[Gal4]) were combined in vivo, with isoform specific UAS lines (called UAS-PGRP-LB[PA], [PC] and [PD]) in a *PGRP-LB* mutant background. Surprisingly, out of the nine possible combinations, only pLB1[Gal4]/UAS-PGRP-LB[PD] was able to fully rescue the egg-laying defect of *PGRP-LB*-infected females (*Figure 2B*). This indicated that all PGRP-LB isoforms are not functionally equivalent as far as egg-laying control is concerned. In order to show that the three isoforms

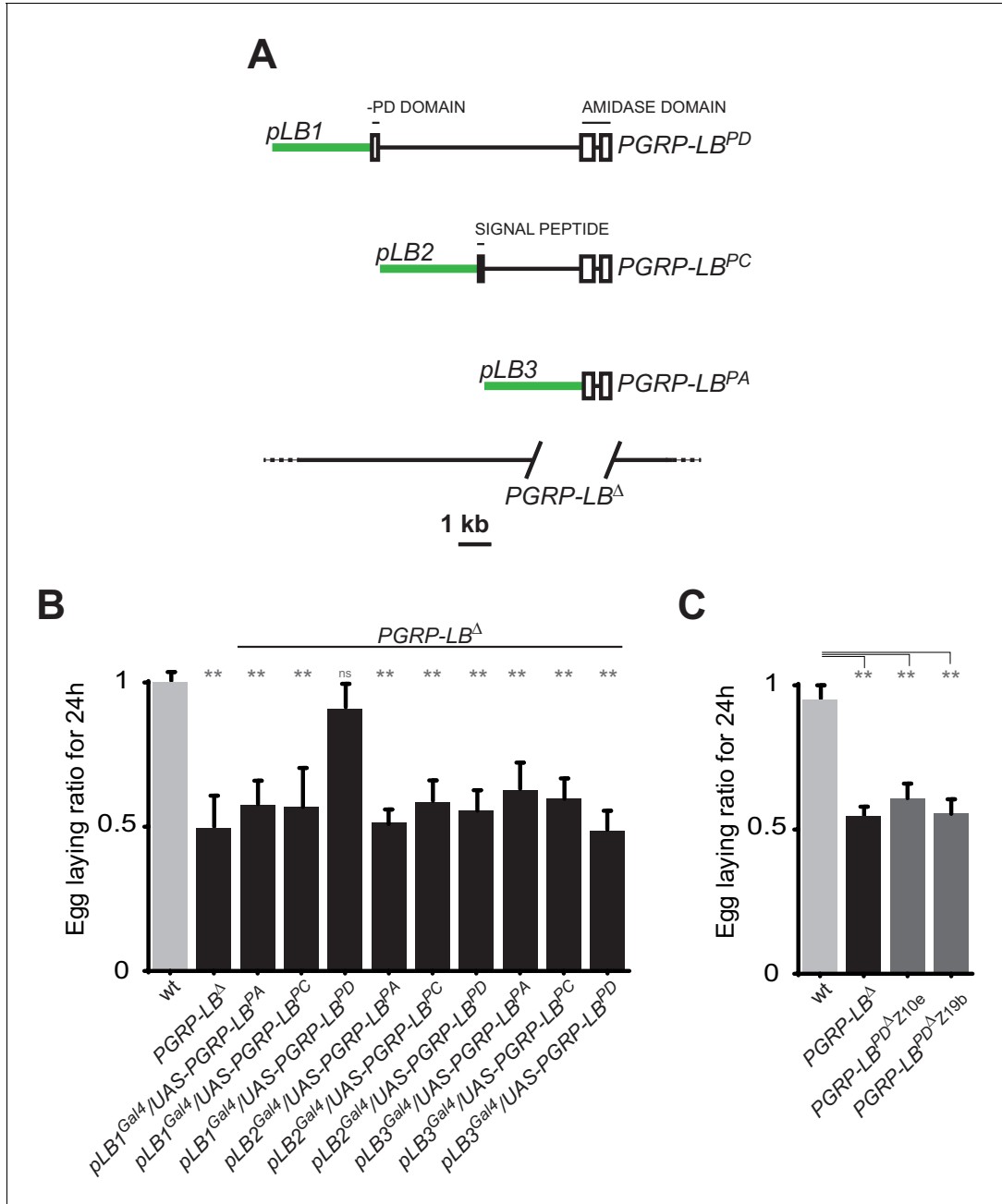

**Figure 2.** One specific PGRP-LB isoform controls oviposition. (**A**) PGRP-LB locus organization. Three PGRP-LB isoforms are produced from the locus. The green lines represent the cloned fragments used to generate the Gal4 constructs. (**B**) Egg-laying ratio of *PGRP-LB* females carrying various Gal4 drivers (pLB1^Gal4, pLB2 ^Gal4, pLB3 ^Gal4) and UAS constructs allowing the overexpression of the three different PGRP-LB isoforms (PGRP-LB^PA, PGRP-LB^PC, PGRP-LB^PD). A single Gal4-UAS combination (pLB1^Gal4/UAS–PGRP-LB^PD) is rescuing the egg-laying drop seen in infected *PGRP-LB* mutant females. (**C**) Egg-laying ratio of *PGRP-LB* mutant females and CRISPR-Cas9-generated *PGRP-LB^PD* only mutants. For **B** and **C**: shown is the average egg-laying ratio ± SEM from at least two independent trials with at least 20 females per genotype and condition used. * indicates $p < 0.01$; ** indicates $p < 0.001$; n.s. indicates $p > 0.05$, unpaired two-tailed Mann Whitney test versus wt animals.

The following figure supplement is available for figure 2:

**Figure supplement 1.** While all PGRP-LB isoforms possess amidase activity, *PGRP-LB^PD* isoform is not required for the negative regulation of the systemic immune response.

retain the ability to cleave PGN, we orally infected *PGRP-LB* mutants that ectopically expressed the different PGRP-LB isoforms in gut cells and monitored the expression of the IMD target gene *Diptericin*. The exacerbated *Diptericin* expression measured in *PGRP-LB* mutant was strongly reduced by expressing any of the three isoforms (*Figure 2—figure supplement 1A*) (*Zaidman-Rémy et al., 2006*). These results suggest that although PGRP-LB isoforms are functionally equivalent in the regulation of gut immune response to bacteria, they have specific roles in regulating bacteria-induced egg-laying phenotype. To further confirm the hypothesis that a single PGRP-LB isoform, PGRP-LB$^{PD}$, was implicated in the control of egg-laying behavior during bacterial infection, we generated via the CRISPR-Cas9 technology, two PGRP-LB$^{RD}$specific alleles which delete part of the PGRP-LB$^{RD}$ specific exon (*Figure 2—figure supplement 1B*). Although these mutants behave as wild-type flies as far as the immune response is concerned (*Figure 2—figure supplement 1C*), their oviposition phenotype post-infection is identical to that of *PGRP-LB* null females (*Figure 2C*). These results demonstrate that by regulating PGN level, the PGRP-LB enzyme is adjusting both immune response intensity and egg-laying rate to infection levels. However, these two regulations are genetically distinct with egg laying probably controlled intracellularly in the pLB1-positive cells via the PGRP-LB$^{PD}$ isoform and immunity extracellularly by the PGRP-LB$^{PC}$ isoform.

## NF-κB pathway activation in pLB1 cells is required to modulate egg-laying post-infection

The results presented above suggest that PGRP-LB$^{PD}$ acts in pLB1 cells to prevent egg-laying reduction in infected *PGRP-LB* mutant females. To test if the effects of bacteria on egg-laying phenotype are a consequence of IMD pathway overactivation in this subset of cells, we blocked IMD pathway specifically in pLB1 cells. RNAi-mediated inactivation of either Relish or PGRP-LE in pLB1 cells was indeed sufficient to block the effect of bacteria on female egg-laying behavior (*Figure 3A*). Similar effects using RNAi-mediated inactivation of Fadd were observed in the *PGRP-LB* mutant background (*Figure 3B*). These results demonstrate that IMD pathway activation via PGRP-LE in pLB1 cells is responsible for the oviposition decrease post-infection.

## NF-κB acts in neurons to regulate infection-dependent egg laying

To elucidate how IMD pathway activation in pLB1 cells would block egg laying, we analyzed the expression pattern of a pLB1$^{QF}$/QUAS-GFP reporter line. GFP staining was detected in neuronal-like fibers in the lateral and common oviducts where it seems to connect the oviduct to the basal part of the ovaries (*Figure 4A'''*). Projections were also specifically detected in the posterior part of the ventral nerve cord, in the abdominal ganglion and in the subesophageal ganglion of the brain (*Figure 4A' and A''*). These results suggest that the effect of PGN on oviposition could be a consequence of NF−κB pathway activation in neurons controlling egg-laying behavior in flies. To validate this hypothesis, we first tested whether the exacerbated reduction of egg deposition in infected *PGRP-LB* mutant females could be rescued by providing PGRP-LB$^{PD}$ in neurons only. *PGRP-LB* females in which the UAS-PGRP-LB$^{PD}$ construct was driven by the pan-neuronal Gal4 driver (elav$^{Gal4}$) no longer show a drop in egg-laying post-infection (*Figure 5A*). In addition, RNAi-mediated inactivation of IMD pathway component Relish or Fadd completely suppressed egg-laying drop post-infection (*Figure 5B*). To further confirm the neuronal nature of the pLB1-positive cells controlling egg laying, we expressed proteins able to modulate neuronal activity in pLB1 cells. Specific expression of the Tetanous toxin (TTX) which cleaves synaptobrevin and impairs neurotransmitter secretion in neurons controlling ovulation has been shown to severely perturb oviposition in virgins as well as in mated females (*Soller et al., 2006*). When TTX was expressed in pLB1 cells of mated uninfected females, a strong defect in oviposition was observed (*Figure 5C*). This was not the case when TTX was expressed in pLB3-positive cells. We then tested whether increasing the excitability of the pLB1 cells via overexpression of the TRPA1 ion channel was sufficient to promote female oviposition. Expressing the TRPA1 channel in pLB1, pLB2 and pLB3 cells at the temperature at which TRPA1 is inactive (23°) did not modify egg-laying comportment of mated females (*Figure 5D*). In contrast, pLB1$^{Gal4}$/UAS-TRPA1 females raised at permissive temperature (29°C) laid significantly more eggs than controls (*Figure 5D*). Such an effect of TRPA1 expression on oviposition was not observed when other pLB$^{Gal4}$ drivers were used (*Figure 5D*). Since TRPA1 is expressed in gut entero-endocrine cells were it mediates the effect of bacteria cell wall component, lipopolysaccharide, we tested

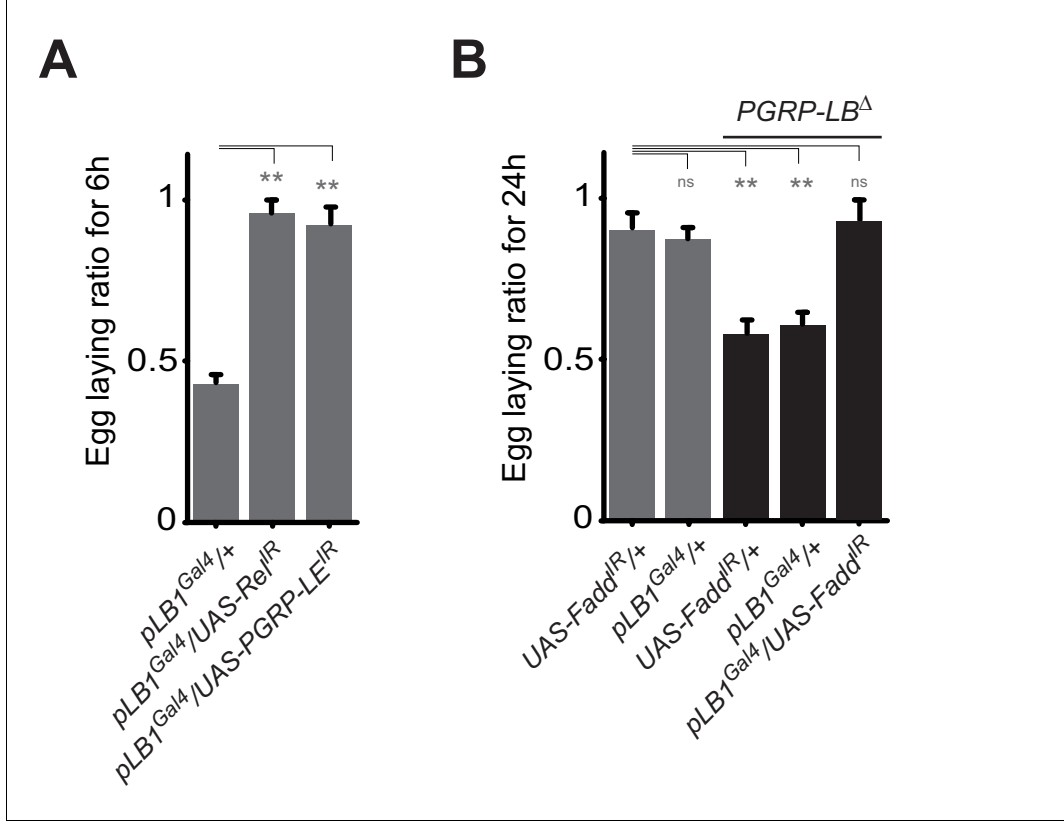

**Figure 3.** A functional NF-κB pathway is required in pLB1 cells to modulate egg-laying post-infection. (**A**) Egg-laying ratio of wt females in which the IMD pathway has been inactivated in pLB1 cells only (pLB1^Gal4/UAS-Rel^IR and pLB1^Gal4/UAS-PGRP-LE^IR)(**B**) Egg-laying ratio of *PGRP-LB* mutant females in which the IMD pathway has been inactivated in pLB1 cells only (pLB1^Gal4/UAS-Fadd^IR). For **A** and **B**; shown is the average egg-laying ratio ± SEM from at least two independent trials with at least 20 females per genotype and condition used. * indicates p<0.01; ** indicates p<0.001; n.s. indicates p>0.05, unpaired two-tailed Mann-Whitney test versus controls.

if these specific gut cells are required for egg-laying response to infection. NF-κB pathway inactivation using two entero-endocrine cells drivers, voila-Gal4 and prospero-Gal4, did not interfere with egg-laying behavior (*Figure 5—figure supplement 1*). Taken together, these results demonstrate that pLB1 cells activity impacts oviposition and that in neurons, the IMD pathway is modulated by the PGRP-LB^PD isoform to fine tune the oviposition drop following exposure to bacteria. These data provide a functional link between the neuronal networks that control oviposition and its regulation by the presence of bacteria.

## Bacterial infection modulates egg laying by interfering with the octopamine pathway

To identify the type of neuron involved, we tested the consequence of NF-κB inactivation in two categories of cells previously implicated in controlling egg-laying behavior that are the ILP7 and ppk-positive neurons (*Castellanos et al., 2013*; *Rezával et al., 2012*; *Lee et al., 2016*). However, as shown *Figure 5—figure supplement 1A*, Fadd RNAi overexpression in these neurons did not prevent egg-laying drop post-infection. In addition, overexpression of the SPR cDNA under the pLB1^Gal4 driver was not sufficient to rescue egg-laying behavior of SPR null mutants (*Figure 5—figure supplement 1B*). Further insights came from looking at the phenotype of unfertilized females. Egg laying in mated females is mainly controlled by signals present in the sperm stored post-mating and the phenotype we observed could be the result of a regulation of this post-mating program (*Kubli and Bopp, 2012*). However, virgin females also produce unfertilized mature oocytes that they released regularly. Importantly, using *PGRP-LB* mutants, we noticed that bacterial infection was also

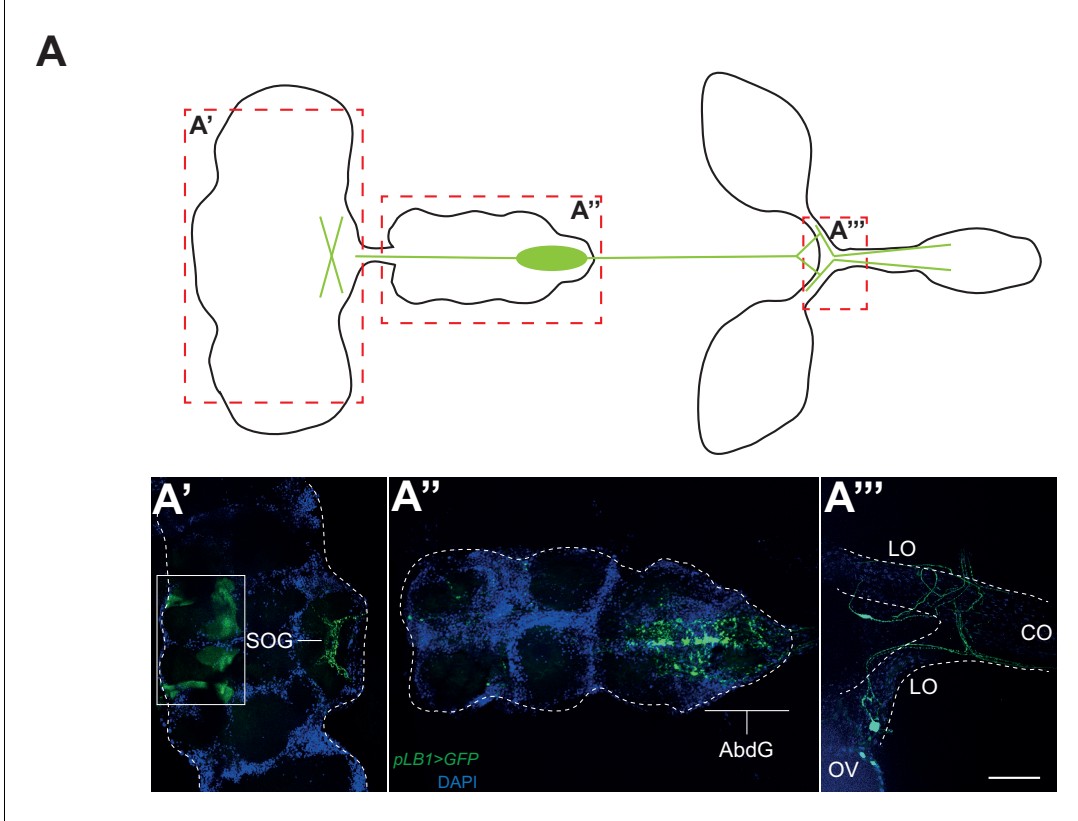

**Figure 4.** pLB1 is expressed in cells of the nervous system. (**A**) Diagram summarizing pLB1 expression pattern (green lines) in the brain (**A'**), the Ventral Nerve Cord (VNC) (**A''**) and the ovaries (**A'''**). Expression pattern of pLB1$^{QF}$/QUAS-GFP (pLB1>GFP) in females. In the brain (**A'**), pLB1 is expressed in the Sub Oesophageal Ganglion (SOG). The area in the box corresponds to unspecific transgene expression seen in the QUAS-GFP strain alone. In **A''**, the expression is restricted to the Abdominal Ganglia (AbdG) that corresponds to the posterior tip of the VNC. In **A'''**, lateral oviducts (LO) connecting ovaries (OV) to the central oviduct (CO) are shown. Staining is also observed in other tissues such as pericardiac cells, some enterocytes (data not shown). The scale bar corresponds to 50 μm.

affecting egg-laying behavior of virgin females (**Figure 6A**). Moreover, we could show that inactivation of pLB1, but not of pLB3 cells via TTX overexpression impacted wild-type virgin females oviposition rate (**Figure 6B**). Similar results with virgins were obtained using the inwardly-rectifying K+ channel Kir (2.1b) that inhibits neuronal activity (**Figure 6—figure supplement 1A**). pLB1$^{Gal4}$/UAS-TTX virgin females presented ovaries packed with stalled mature oocytes which led to an important increase of ovaries size and, in turn, of female abdomen (**Figure 6—figure supplement 1B and C**). Both phenotypes are reminiscent of those seen in mutants for the octopamine pathway ($T\beta H^{nM18}$, **Figure 6—figure supplement 1C**) or in flies in which octopaminergic neurons are functionally inactivated (**Monastirioti et al., 1995**). The monoamine octopamine functions as a key neurotransmitter for ovulation in flies. It is synthesized from tyrosine by the sequential actions of Tyrosine decarboxylase (Tdc2) and Tyramine-$\beta$-hydroxylase (T$\beta$H). Females defective for Tdc2 or for T$\beta$H are sterile and display swollen ovaries where stalled mature eggs are not delivered into the oviducts, mainly due to defects in muscle contraction (**Schwaerzel et al., 2003**). Taken together, our results suggest that bacterial infection could trigger an IMD pathway activation in octopaminergic neurons that in turn would induce egg-laying drop by reducing octopamine signaling. To challenge this hypothesis, we tested whether inactivating the IMD pathway in octopaminergic neurons only, will prevent egg-laying drop after infection. As shown in **Figure 6C and D**, RNAi-mediated NF-κB pathway inactivation in T$\beta$H-positive cells only, was sufficient to prevent the bacterially induced egg-laying drop in wild-type and *PGRP-LB* mutant background. Then, we asked whether NF-κB pathway modulation by PGRP-LB$^{PD}$ was also taking place in the octopaminergic neurons. As shown **Figure 6E**, PGRP-LB$^{PD}$ ectopic expression under the Tdc and the T$\beta$H Gal4 drivers, was sufficient to rescue *PGRP-LB* mutant

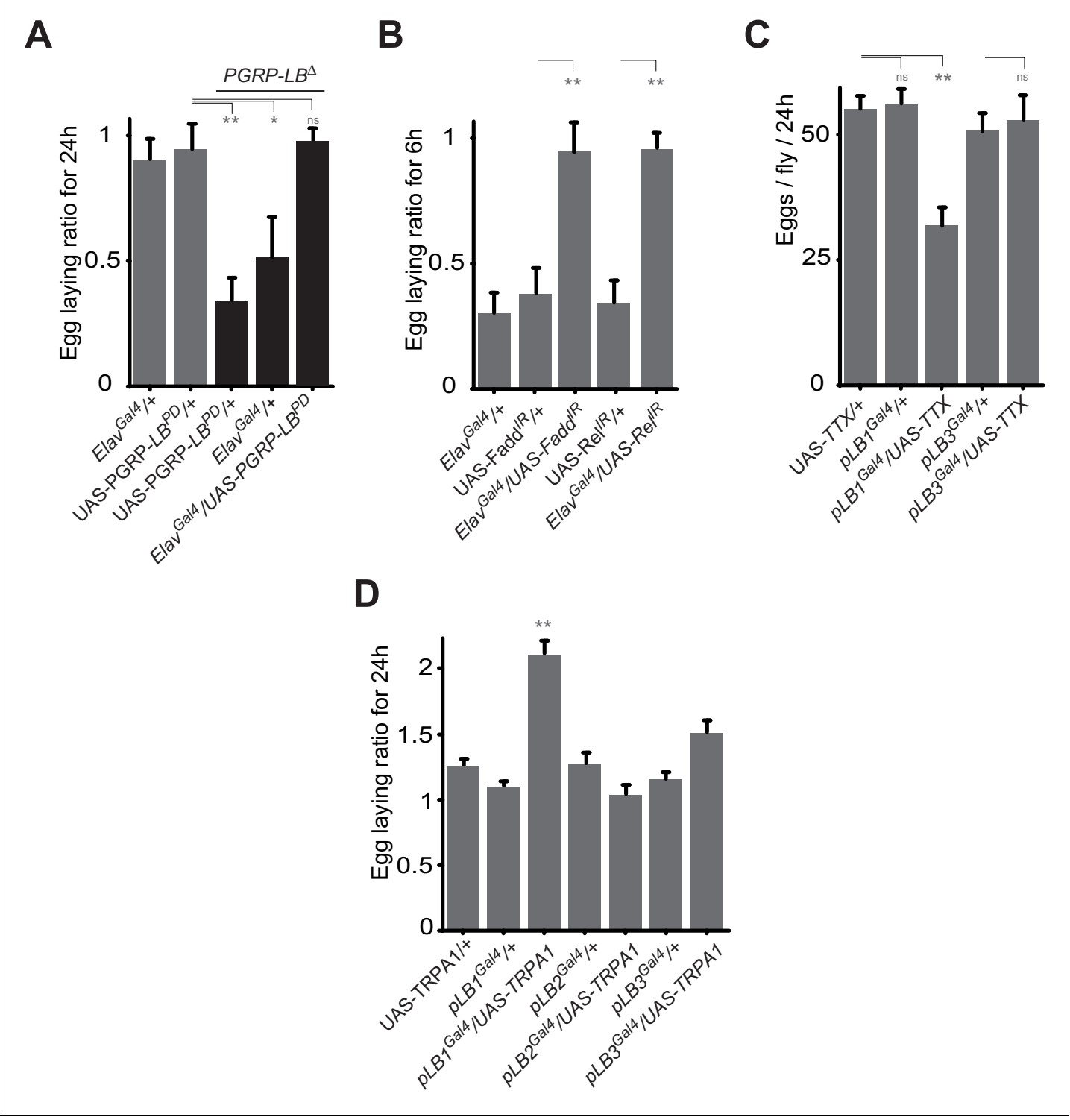

**Figure 5.** Some pLB1 cells are neurons controlling oviposition via NF-κB. (A) Egg-laying ratio of *PGRP-LB* mutant females in which the pan-neuronal Gal4 drivers (Elav $^{Gal4}$) is used to drive expression of the PGRP-LB$^{PD}$ isoform. Expression of the PGRP-LB$^{PD}$ isoform in neurons only is sufficient to rescue *PGRP-LB* mutant phenotype following septic injury. (B) RNAi-mediated inactivation of the NF-κB pathway prevents egg-laying drop post-infection. (C) Expression of Tetanus Toxin (TTX) in pLB1, but not in pLB3-positive cells, is sufficient to decrease egg laying in non-infected mated females. *pLB2$^{Gal4}$/ UAS-TTX* flies are not viable (data not shown). (D) Expression of the transient receptor potential cation channel (TRPA1) in pLB1 but neither in pLB2 nor pLB3-positive cells stimulates egg laying in non-infected mated females. For **A** and **B**; shown is the average egg-laying ratio ± SEM from at least two independent trials with at least 20 females per genotype and condition used. For **C**; shown is the average number of eggs laid per fly per 24 hr ± SEM
*Figure 5 continued on next page*

*Figure 5 continued*

from at least two independent trials with at least 20 females per genotype and condition used. For **D**, shown is the average egg-laying ratio, the number of eggs laid by females raised at 29°C (permissive temperature for TPRA1) over the number of eggs laid at 23°C (restrictive), ± SEM from at least two independent trials with at least 20 females per genotype and condition used. * indicates p<0.01; ** indicates p<0.001; n.s. indicates p>0.05, unpaired two-tailed Mann-Whitney test versus controls (for **A**, **B** and **C**) and Dunn's multiple comparison test (for **D**).

The following figure supplement is available for figure 5:

**Figure supplement 1.** *ppk*, *Ilp7*, *prospero* or *voila*-positive cells do no regulate female egg laying after infection.

egg-laying phenotype. Finally, we tested whether the regulation of the oviposition following exposure to bacteria was modulated via the quantity of octopamine. Thus, we increased the amount of octopamine via T$\beta$HcDNA overexpression in pLB1 cells and measured the egg-laying drop phenotype of infected *PGRP-LB* mutants. Providing an excess of T$\beta$H in pLB1, but neither in pLB2 nor in pLB3 cells, was sufficient to fully rescue *PGRP-LB* oviposition drop following infection (*Figure 6F*). Altogether these results indicate that bacterial infection induces in neurons a NF-κB-pathway-dependent regulation of octopaminergic signal which in turns modulates ovulation and triggers a reduction of female oviposition. These effects are fine-tuned in the octopaminergic neurons by one specific PGRP-LB isoform, PGRP-LB[PD] (*Figure 7*).

## Discussion

Our present data demonstrate that bacteria derived PGN entry into the fly body cavity has, at least, two physiological consequences (*Figure 7*). In addition to activate innate immune response in fat body cells, it also blocks mature egg delivery in oviduct and hence reduces egg laying of infected females. We further demonstrate that this bacterially induced behavioral change is due to an NF-κB pathway-dependent modulation in octopaminergic neurons. We finally present evidence that both responses, that are potentially detrimental if not down-regulated, are fine-tuned by distinct and specific PGN degrading enzymes. We propose that by regulating the level of internal PGN, flies adapt their egg-laying behavior to environmental conditions. In standard environmental conditions, PGRP-LB ensures that low level of PGN does not affect egg laying. However, whenever PGN concentration reaches a certain threshold, which either reflects an infection status or the presence of a highly contaminated food supply, NF-κB pathway activation in neurons is blocking egg release. As PGN of ingested bacteria is capable of reaching the internal fluid and triggering dedicated signaling cascades (*Basset et al., 2000*), one could imagine that such a mechanism prevents flies to lay their eggs in highly contaminated food in which their development and that of the hatching larvae could be impaired by microbes. In this context, PGRP-LB mediated PGN scavenging is crucial since a non-regulated behavioral immune response would lead to a severe drop in the amount of progeny which may not be in keeping with the real threat. Another possibility could be that a reduced egg production will favor immune effector production. Indeed, it is often considered that the energy cost of an acute innate immune response needs can be balanced by a decreased offspring production (*Stahlschmidt and Adamo, 2013*). Blocking the energy-consuming egg production in infected flies could be a way for them to mobilize resources required for full activation of innate immune defences. A similar depression of oviposition has recently been documented in females flies exposed to parasitoid wasps who lay their eggs in *Drosophila* larvae (*Kacsoh et al., 2015*). However, while visual perception of wasps by female flies induces a long-term decline in oviposition associated with an early stage-specific oocyte apoptosis, PGN effects are transient and rather lead to a late stage oocyte accumulation suggesting that although the final outcome is the same, the mechanisms differ.

The present data indicate that PGN sensing acts on egg-laying behavior via neuronal modulation. We identified NF-κB pathway signaling in octopaminergic neurons as the actor of this PGN-dependent oviposition reduction. It would be informative to test whether bacterial infection is also affecting other octopamine-mediated behaviors such as reward in olfactory or visual learning, male-male courtship, male aggressive behavior (*Unoki et al., 2006*; *Certel et al., 2007*; *Zhou et al., 2008*) or sleep:wake regulation (*Crocker et al., 2010*). This would require to further characterize the nature of the octopamine neurons whose activation is modulated by infection and to consider that the

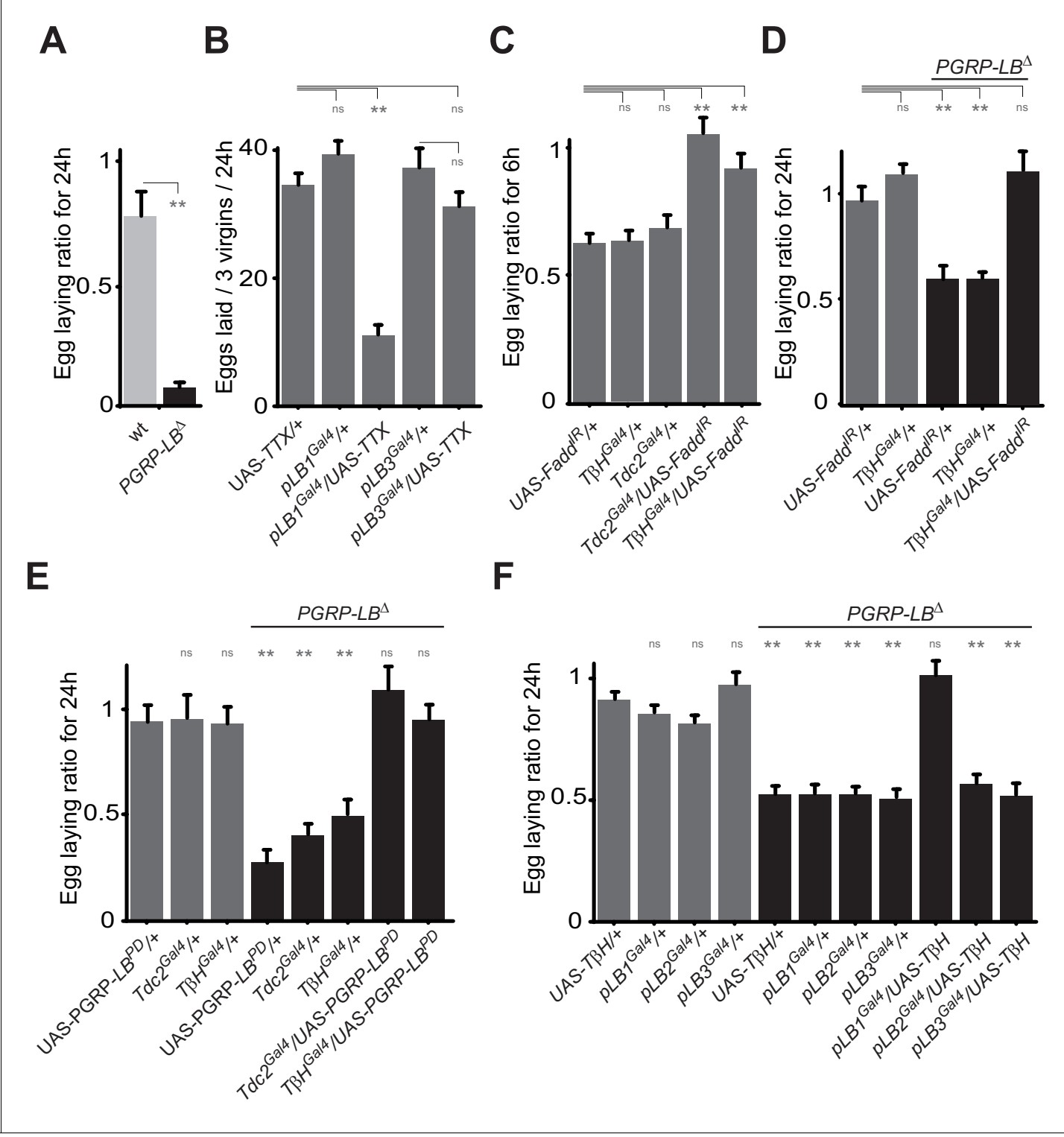

**Figure 6.** Bacteria modulate egg-laying behavior via the octopamine pathway. (**A**) Septic injury reduces egg laying in *PGRP-LB* virgin females. Egg-laying ratio post-septic injury of wt and *PGRP-LB* mutant virgin females. (**B**) Functional inactivation of pLB1, but not of pLB3-positive cells, by UAS-TTX blocks egg laying in virgin females. Total eggs laid by wild-type virgins in which the TTX is expressed in pLB1 or pLB3 cells. (**C** and **D**) Egg-laying ratio of wt (**C**) or *PGRP-LB* mutant mated females (**D**) in which the IMD pathway has been specifically inactivated via UAS-Fadd[IR] ectopic expression in TβH or Tdc2-positive cells. (**E**) Restoring PGRP-LB[PD] expression in cells that produce the enzymes required to synthesize octopamine (tdc2 and TβH) fully rescues the *PGRP-LB* mutant phenotype following septic injury. Egg-laying ratio of *PGRP-LB* mutant females in which the tdc2 [Gal4] and TβH[Gal4] drivers are used to overexpress the PGRP-LB[PD] isoform. (**F**) Providing an excess of TβH in pLB1 cells is sufficient to rescue *PGRP-LB* mutant phenotypes. Egg-

*Figure 6 continued on next page*

*Figure 6 continued*
laying ratio of *PGRP-LB* mutant mated females in which the TβH level has been increased in pLB1, pLB2 or pLB3 cells. For **A**, **C**, **D**, **E** and **F**; shown is the average egg-laying ratio ± SEM from at least two independent trials with at least 20 females per genotype and condition used. For (**B**); shown is the average number of eggs laid per three virgins per 24 hr ± SEM from at least two independent trials with at least 20 females per genotype and condition used. * indicates p<0.01; ** indicates p<0.001; n.s. indicates p>0.05, unpaired two-tailed Mann-Whitney test versus indicated controls for **A**, **B**, **C** and **D** and versus UAS in wt background for **E** and **F**.
The following figure supplement is available for figure 6:

**Figure supplement 1.** Inactivation of pLB1 cells phenocopies *TβH* mutant.

phenotypes defined as being part of the sickness behaviours might be orchestrated directly by the immune system following the perception of microbes. Indeed, as shown above, our PGRP-LB^PD reporter line is not only labeling cells in the reproductive tract but also in thoraco-abdominal ganglia and in the brain with projections to proboscis, wings and legs. Likewise, octopaminergic neurons have been shown to innervate numerous areas in the brain and in the thoraco-abdominal ganglion and to project to various reproductive structures such as ovaries, oviducts and uterus, further work will be needed to exactly pinpoint the identity of the affected octopaminergic neurons, their targets and their effect on fly behavior. In addition, the question remains as to how NF-κB activation can modulate octopaminergic neurons activity. Among the possibilities is the modulation of octopamine neuron excitability, the regulation of octopamine production or its secretion. Knowing the NF-κB protein itself is required for this behavioral response and that increasing the amount of available octopamine via overexpression of the TβH enzyme rescues the oviposition drop, it is expected that IMD pathway activation in neurons will have transcriptional consequences. However, other hypotheses might be considered since Dorsal, a member of the other *Drosophila* NF-κB signaling cascade Toll, has been shown to function post-transcriptionally together with IκB and IRAK at the post-

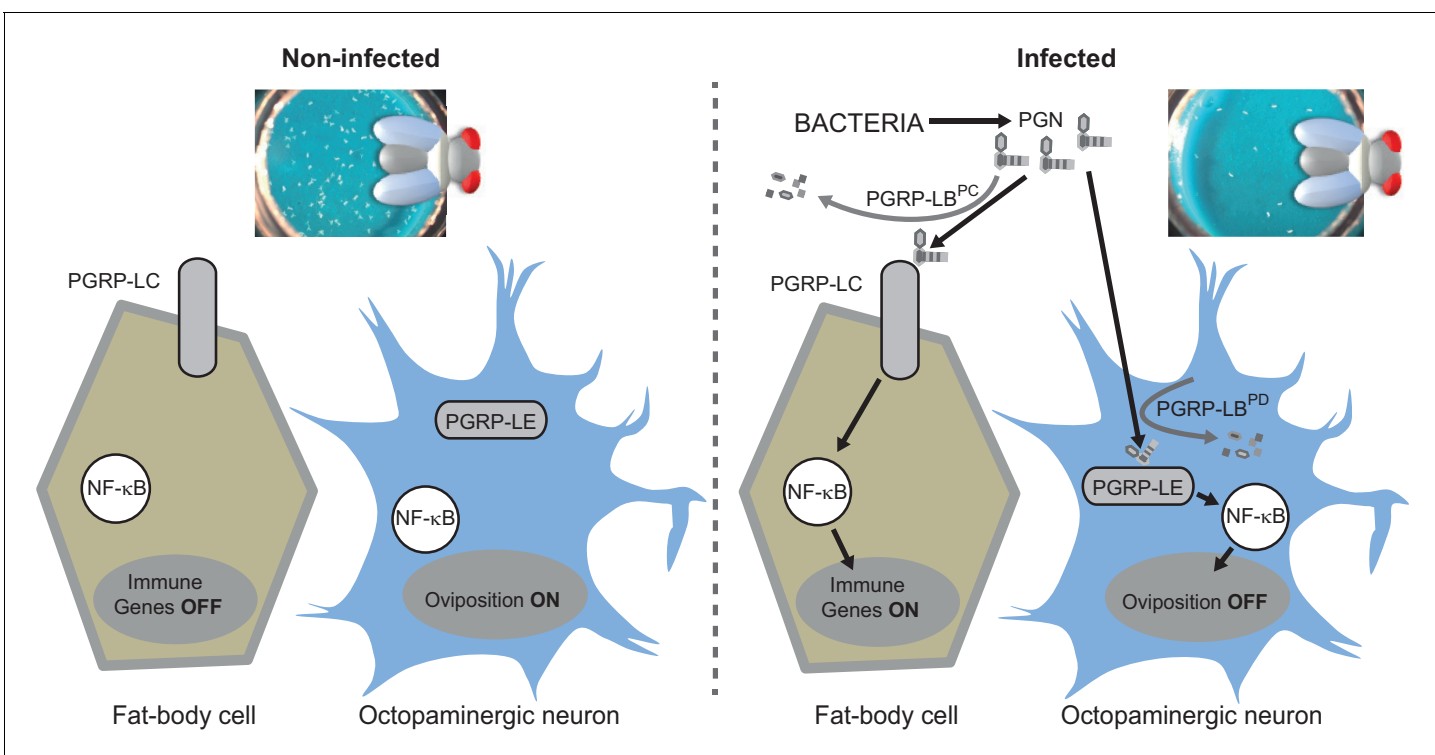

**Figure 7.** Diagram summarizing the impact of PGN sensing on the expression of immune genes in fat body cells as well as on the regulation of oviposition in octopaminergic neurons.

synaptic membrane to specify glutamate receptor density (*Heckscher et al., 2007*). It should also be noticed that PGRP-LC has recently been shown to control presynaptic homeostatic plasticity in mouse (*Harris et al., 2015*). One of the future challenges will be to understand how NF-κB activation is reducing octopaminergic signals.

We show here that *Drosophila* uses an unique bacteria associated molecular pattern to activate different processes related to host defence, namely the production of antimicrobial peptides and the modulation of oviposition behavior. Interestingly, it appears that in order to fine-tune these responses, different isoforms of the same PGN scavenging enzyme, PGRP-LB, are required. While the secreted PGRP-LB^PC isoform certainly acts non cell-autonomously to dampen immune activation by circulating PGN (*Zaidman-Rémy et al., 2006*), a putatively intracellular isoform PGRP-LB^PD controls the effect of PGN on oviposition. Even more remarkable, this response is not transmitted via PGRP-LC but rather by the intracytoplasmic PGRP-LE receptor. Previous work has shown that PGRP-LE is also regulating response to bacteria in some part of the gut (*Bosco-Drayon et al., 2012*; *Neyen et al., 2012*). Thus, it will be important to understand how PGN is trafficking within and through cells, and how PGRP-LB^PD modulates PGRP-LE-dependent IMD pathway activation and whether it is also required to modulate other PGN/PGRP-LE-dependent responses.

In essence, our results demonstrate that PGN, when ingested or introduced into the body cavity, not only activates antibacterial immune response but also influences neuronally controlled behaviors in flies. Importantly, the sickness behavior we deciphered does not appear to be a side effect of an energetically expensive immune response, but rather the result of a specific regulation. An orchestration of different processes required for the immune response was also exemplified by a recent report linking metabolism and immunity (*Clark et al., 2013*). Although not dissected to the molecular level, previous studies in mammals have suggested that similar interactions between PGN and neuronally controlled activities. For instance, PGN derived muropeptide MDP has been shown to display powerful somnogenic effect when injected into rabbit brain (*Krueger et al., 1984*). It has also been shown that PGN produced by symbiotic microbiota may 'leak' into the bloodstream and reach organs distant to the gut, such as the bones (*Clarke et al., 2010*). Finally, recent findings show that bacterial cell wall peptidoglycan traverses the murine placenta and reach the developing fetal brain where it triggers a TLR2-dependent fetal neuroproliferative response (*Humann et al., 2016*). A future challenge will be to test whether an NF-κB-dependent response to PGN is also taking place in mammalian neurons and directly influences the animal behavior.

## Material and methods

### Bacteria

The bacterial strains used are *Escherichia coli* strain DH5α (grown at 37°C) and *Erwinia carotovora carotovora 15 2141* (*Ecc*, grown at 30°C). Strains were cultured in Luria-Bertani medium with shaker overnight. Bacterial cultures were centrifuged at 2500 g for 15 min at room temperature (RT) and re-suspended in fresh Luria-Bertani medium. OD at 600 nm was measured and bacteria were diluted in LB medium to the desired concentration.

### *Drosophila melanogaster* strains and maintenance

The following strains were used in this work: PGRP-LE^112 (*Kaneko et al., 2004*), PGRP-LC^ΔE12 (*Gottar et al., 2002*), Dredd^D55 (*Leulier et al., 2000*), Relish^E20 (*Hedengren et al., 1999*), pirk^EY0073 (*Lhocine et al., 2008*), PGRP-LB^Δ (*Paredes et al., 2011*), Df(3R)^ED5516 (BL#8968), Df(1)Excel^6234 and TβH^nM18 (kindly provided by Henrike Scholz). UAS-nlsGFP (BL#4775), UAS-IMD (*Georgel et al., 2001*), UAS-TTX (*Sweeney et al., 1995*), UAS-TRPA1 (*Hamada et al., 2008*), UAS-Kir2.1 (*Hardie et al., 2001*), QUAS-GFP (BL# 52264), UAS-TβH (kindly provided by Henrike Scholz) and UAS-Fadd^IR, (*Khush et al., 2002*) (Kindly provided by P. Meier), UAS-Rel^IR (BL#28943) and UAS-LE^IR (BL#60038). Elav-Gal4 (BL# 8760), NP1-Gal4 (DGRC-Kyoto #112001), Tdc2-Gal4 and TβH-Gal4 (kindly provided by Henrike Scholz), ppk-Gal4 (kindly provided by Barry Dickson), ILp7-Gal4 (Kindly provided by Irene Miguel-Aliaga), prospero-Gal4 and Voila-Gal4 (kindly provided by Armel Gallet).

Flies were grown at 25°C on a yeast/cornmeal medium in 12 hr/12 hr light/dark cycle controlled incubators. For 1 L of food, 8.2g of agar (VWR, cat. #20768.361), 80g of cornmeal flour (Westhove, Farigel maize H1) and 80 g of yeast extract (VWR, cat.#24979.413) were cooked for 10 min in boiling

water. 5.2 g of Methylparaben sodium salt (MERCK, cat.#106756) and 4 ml of 99% propionic acid (CARLOERBA, cat. #409553) were added when the food had cooled down. For antibiotic treatment (ATB), standard medium was supplemented with Ampicillin, Kanamycin, Tetracyclin and Erythromycin at 50 µg/ml final concentrations.

## Transgenic lines generated
UAS-PGRP-LB[PA], UAS-PGRP-LB[PC], UAS-PGRP-LB[PD], pLB1[Gal4], pLB2[Gal4], pLB3[Gal4], and pLB1QF. To generate the UAS-PGRP-LB[PA] and UAS-PGRP-LB[PC] lines, a DNA fragment corresponding to their respective ORF was PCR amplified from the GH21008 and the RE34140 DGRC EST clones. To construct the UAS-PGRP-LB[PD], we first performed two PCR reactions using GH21008 and genomic DNA as template for common ORF and RD specific exon, respectively. The two resulting PCR products were then combined to generate the PGRP-LB[PD] ORF amplicon. Using Gateway Technology, the PGRP-LB[RA], PGRP-LB[RC] and PGRP-LB[RD] fragments were cloned into the pTWG vector (C-terminal eGFP tag with a UAS promoter). To construct the pLB1[Gal4], pLB2[Gal4] and pLB3[Gal4] lines, a DNA fragment corresponding to the respective determined promoter was PCR amplified from the BAC CH321-82G09. Using Gateway Technology, the pLB1, pLB2 and pLB3 fragments were cloned into the SMG4-Gal4 vector. For the pLB1[QF] line, the previously mentioned pLB1 PCR fragment was cloned into pattB-QF-hsp70 vector from Addgene (Plasmid #24368). Molecular details are available upon request.

## Generation of CRISPR alleles
### Generation of the y1, w1118 P(3xP3-EGFP, vasa-cas9D10A)attP18 stock (by Kate Koles and Avi Roda at Brandeis University, USA)
The miniwhite marker in pBID-G (a gift from Brian McCabe; Addgene plasmid # 35195) was replaced by a 3xP3-EGFP cassette using the ApaI and NheI restriction sites, yielding pBID-3xP3-EGFP. 3xP3-EGFP was PCR amplified from pBSII-SK-3xP3-EGFP-vasa-dPhiC31attB (a gift of Drs. Johannes Bischof and Konrad Basler). The 3xP3-EGFP cassette drives strong EGFP expression under the artificial Pax-6/eyeless-derived promoter element in the nervous system. The HA-NLS-Cas9[D10A]-NLS sequence was PCR amplified from pX334-U6-DR-BB-DR-Cbh-NLS-hSpCas9n(D10A)-NLS-H1-short-tracr-PGK-puro (a gift from Feng Zhang, Addgene plasmid # 42333) and TOPO cloned to pENTR-D-TOPO vector. Subsequently, the 5' and 3' *vasa* gene regulatory elements (*Bischof et al., 2007*) were PCR amplified from pBSII-SK-3xP3-EGFP-vasa- dPhiC31attB and cloned into pENTR-D-TOPO-Cas9 [D10A] vector yielding pENTR-D-TOPO-vasa5'-Cas9[D10A]-vasa3'. This was then Gateway LR cloned into pBID-3xP3-EGFP, yielding the final plasmid pBID-3xP3-EGFP-vasa-Cas9[D10A]. Rainbowgene was used to create the transgenic flies at docking site attP18Plasmid encoding two synthetic guide RNAs (sgRNAs) was injected (100 ng/µl) into preblastoderm embryos of *nos*PhiC31; attP40 flies. sgRNAs (sgRNA1: AGTTGGGCCAGAAGTGGAGA and sgRNA2: CCTCATGGACTCGGAGCAAT) were designed using the online CRISPR design tool (http://crispor.tefor.net) to target RD specific exon of PGRP-LB and were cloned into the pCFD4-U6:1_U6:3 vector (Addgene #49411; [*Port et al., 2014*]). Males transgenic flies were crossed with Cas9 nickase transgenic females flies. This strategy reduces off-target activity. We used the y1, w1118 P(3xP3-EGFP, vasa-cas9D10A)attP18 fly line . G0 founder flies were crossed with w1118;;MKRS/TM6b flies then the F1 progeny was screened for mutations (indels or larger deletions) by high resolution melting analysis.

## Oviposition assays
In order to ease the quantification of the laid eggs, a blue food dye (E133, Le meilleur du chef) was incorporated (1%) into the antibiotic media used for the oviposition assays (Blue-ATB). The egg-laying index corresponds to the ratio between the number of eggs laid by a treated female and the average number of eggs per tube laid by the untreated animal during a specific period of time, 24 hr when not otherwise stated. An oviposition ratio of 1 indicates that the treatment did not impact the oviposition of the tested female during the time course of the experiment. *Septic injury of mated females*: One-day-old animals were harvested from ATB tubes kept at 25°C. Males and females were mixed in one tube with no more than 40 individuals per tube. Tubes were kept at 25°C and flies shifted to fresh tubes every 2 days. On day 5, females were used for septic injury. All the flies including control animals were anesthetized on $CO_2$ pad. Untreated and treated animals were then

transferred, one per fresh Blue-ATB tube and stored 24 hr at 25°C. Septic injuries were always performed between ZT0 and ZT6 with a tungsten needle dipped into a fresh solution of bacteria at OD200. *PGN injections*: As for septic injuries except that pure PGN was used and injected using a nanojector (Nanojet II, Drummond Scientific Company, PA, USA). PGN is from *E. coli* (Invivogen, ref 14C14-MM, CA, USA) and was resuspended in endotoxin-free water at 200 µg/mL. 9 nL of PGN solution was injected in the thorax. *Feeding with bacteria*: Five-day-old flies were pooled on 2% sucrose/LB/bacteria OD200 or 2% sucrose/LB in tubes without media containing at the bottom cotton filled with 1 mL of solution. Animals were kept 24 hr at 25°C in these tubes then individually shifted to ATB tubes. Eggs were counted for each tube 24 hr later. *Assays with virgins*: Virgins were harvested from ATB tubes kept at 25°C. Tubes were kept at 25°C, flies shifted in new tubes every 2 days. Five- to seven-day-old individuals were used for septic injury as described above except that 3 to 4 virgins were used per tube. Eggs were counted 24 hr later.

## Quantitative real-time PCR

mRNA from whole adults or dissected organs (n = 30) was extracted with RNeasy Mini Kit (QIAGEN, cat. #74106, Germany). Quantitative real-time PCR, TaqMan, and SYBR Green analysis were performed as previously described [36]. Primers information can be obtained upon request. The amount of mRNA detected was normalized to control rp49 mRNA values. Normalized data were used to quantify the relative levels of a given mRNA according to cycling threshold analysis (ΔCt).

## Ovaries content

Flies were reared and harvested as for the oviposition assays except that 10 treated or untreated females were pooled by ATB tubes. At specific time points, after a 20 s EtOH bath, animals were dissected at RT in 1XPBS then ovaries gently opened on a glass slide in a 1XPBS drop. Four categories of stages from 8 to 14 were visually quantified per ovary using a Leica MZ6 binocular scope (Germany). At least 10 ovaries per conditions were used.

## DCP1 immunostaining

Flies were reared and harvested as for the oviposition assays. After 6 hr, ovaries were dissected and fixed 1 hr in 4% paraformaldehyde. Anti-DCP1 (Cell signaling Technology , #9578) primary antibody was used.

## Food-intake assay

Flies were reared and harvested as for the oviposition assays except that after 6 hr, guts and crops were dissected in cold PBS. Five guts and crops were pooled and transferred to 1.5 mL canonical tubes (VWR #16466–064) containing 0,75–1 mm glass beads (Retsch #22.222.0004, Germany) and 50 µL PBS and ground automatically using a grinder (Precellys 24, France). The lysates were then centrifuged at 15000 rpm and the OD 630 nm was measured using a Nanodrop (Thermo Scientific NanoDrop 1000, DE, USA).

## Imaging

Adult tissues were dissected in PBS, fixed for 20 min in 4% paraformaldehyde on ice and rinsed three times in PBT (PBS + 0.1% Triton X-100). The tissues were mounted in Vectashield (Vector Laboratories, Ca, USA) fluorescent mounting medium, with or without DAPI. Images were captured with either a Zeiss Stereo Discovery V12 microscope or an AxioImager APO Z1 apotome microscope (Zeiss, Germany).

## Statistical analyses and graphics

The Prism software (GraphPad) was used for statistical analyses. Our sets of data were tested for normality using the D'Agostino-Pearson omnibus test, and some of our data did not pass the normality test. Consequently, we used non-parametric tests for all the data sets and mainly the unpaired Mann-Whitney two-tailed test. Moreover, we do not show one experiment representative of the different biological replicates, but all the data generated during the independent experiments in one graph.

## Acknowledgements

We thank Barry Dickson, Armel Gallet, Bruno Lemaitre, Pascal Meier, Ryu Ueda, Sadia Deddouche, Kate Koles, Avi Roda, Henrike Scholz, for fly stocks. We thank Cedric Maurange, Marie Meister, Benjamin Prud'homme, Thomas Riemensperger, Thomas Rival and Olivier Zugasti for comments on the manuscript. Imaging were performed on PiCSL-FBI core facility (IBDM, AMU-Marseille) supported by the French National Research Agency through the « Investments for the Future' program (France-BioImaging, ANR-10-INBS-04). This work was supported by CNRS, Equipe FRM to Julien Royet 'Equipe FRM DEQ20140329541', ANR-11-LABX-0054 (Investissements d'Avenir–Labex INFORM).

## Additional information

### Funding

| Funder | Grant reference number | Author |
|---|---|---|
| Centre National de la Recherche Scientifique | 24567 | Julien Royet |
| Equipe Fondation pour la Recherche Médicale | DEQ20140329541 | Julien Royet |
| Investissements d'avenir-Labex INFORM | ANR-11-LABx-0054 | Julien Royet |

The funders had no role in study design, data collection and interpretation, or the decision to submit the work for publication.

### Author contributions

CLK, Conceptualization, Data curation, Methodology, Writing—original draft; BC, Conceptualization, Data curation, Investigation, Methodology, Writing—original draft; DC, Data curation, Formal analysis; AV-L, Data curation, Formal analysis, Investigation; JR, Conceptualization, Data curation, Formal analysis, Funding acquisition, Investigation, Writing—original draft, Project administration, Writing—review and editing

### Author ORCIDs

Bernard Charroux, http://orcid.org/0000-0002-3359-4066
Julien Royet, http://orcid.org/0000-0002-5671-4833

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
