## [Decision Letter]

Thank you for submitting your article "Peptidoglycan sensing by octopaminergic neurons modulates *Drosophila* oviposition" for consideration by *eLife*. Your article has been reviewed by two peer reviewers, and the evaluation has been overseen by Mani Ramaswami as the Reviewing Editor and Wendy Garrett as the Senior Editor. The following individual involved in review of your submission has agreed to reveal his identity: Giovanni Bosco (Reviewer #2).

The reviewers have discussed the reviews with one another and the Reviewing Editor has drafted this decision to help you prepare a revised submission.

Summary:

The work explores mechanisms through which *Drosophila* modulate oviposition in response to bacterial infection. An intriguing aspect of this study is that, in addition to physiological responses, *Drosophila* also exhibit behavioral changes in response to bacterial pathogens. The authors suggest this to be an example of a behavioral adaptation that facilitates both protection against the pathogen for the exposed individual as well as affecting fitness by modulating reproduction, e.g. "behavioral immunity". The evidence supports a model whereby octopaminergic neurons act directly on ovarian tissues to modulate egg laying in response to neuronal signaling elicited by bacterial infection. In a series of convincing loss- and gain-of function experiments, the authors establish that bacterially derived peptidoglycan acting through the NF-ΚB (IMD) pathway is responsible for this effect, and describe novel isoform-specificity in the function of the PGN degrading enzyme PGRP-LB in this context. They then present some evidence that the IMD pathway exerts its effects on egg production in a subset of octopaminergic cells, some of which innervate the oviduct.

The text was clear and concisely written, and for the most part figures were easy to follow and data supported the interpretations presented in the text. This study presents a new and exciting link between the innate immune system and the nervous system that allows for a multi-system integration of physiological and behavior responses to relevant environmental challenges, such as infection.

Essential revisions:

In general, it is important to strengthen the genetic experiments concerning neuronal manipulations in order to clarify the neuronal contribution to the described phenotypes.

1) There is some confusion re: the neuronal site of NF-ΚB activation. Based on the CNS neurites in the images provided, the pLB1-expressing cells seem to include peripheral sensory (taste?) neurons and sensory neurons in the oviduct, but not efferent oviduct-innervating neurons such as the octopaminergic neurons, with cell bodies located in the posterior tip of the VNC (see, for example, Monastirioti et al. (2003) Dev Biol PMID 14623230). Puzzlingly, the authors seem to imply that the octopaminergic neurons are a subset of pLB1 neurons – in fact, the subset that matters for the PGN effects on egg laying. To make matters more complicated, some of the drivers used in the study to demonstrate neuronal effects (e.g. elav-Gal4) also target enteroendocrine cells, recently shown to sense luminal bacteria (see Du et al. (2016) PLoS Genet doi: 10.1371/journal.pgen.1005773). While the authors are careful not to ascribe the observed effects to a specific neuronal subset, the paper will be greatly improved by experiments to restrict and better define the specific octopinergic responsible for the phenotype. The authors should use restricted genetic manipulations and detailed neuronal characterisation. For example, using drivers specific for oviduct sensing (ppk) or efferent (Ilp7, oct+) neurons should allow them to establish the whether the IMD pathway acts in sensory, efferent and/or other endocrine cells to mediate the egg laying reduction.

2) Related to this point, the authors need to distinguish between a generic role for neurons in controlling egg laying, relaying the PGN-triggered signal or sensing it. In this regard, I see how presenting egg laying phenotypes as their ratio relative to control conditions makes comparisons visually straightforward, but can also mask "constitutive" phenotypes resulting from some genetic manipulations – especially those resulting from interfering with the activity of oviduct-innervating neurons. The authors should clarify which of their genetic manipulations have a strong effect on the constitutive egg laying ability of flies, and which are specific to septic injury responses.

3) Based on the data provided, it also seems conceivable that the main effect of PGN is to transiently reduce food intake, which can indirectly impact egg production. Do any of the described genetic manipulations affect one phenotype without affecting the other one?

4) The images of ovaries in Figure 1—figure supplement 3 would be more useful if they were stained with a DNA dye that better revealed the mostly translucent very early stage egg chambers, i.e. it's otherwise very difficult to see stage 7/8 and younger egg chamber. Nevertheless, from these few images it is clear that the ovaries 6h post clean injury have vast numbers of very early (stages 2-5/6) whereas 6h post septic injury these early stages are almost completely absent. Because the figure only presents one ovary and the quantitation does not measure earlier stages it is impossible to know what the fate of these earlier stage egg chambers is in response to infection. This is an important point because the authors suggest (Discussion, end of first paragraph) that infection does not trigger oogenesis checkpoints that utilize apoptosis to eliminate early stage oocytes while allowing continued development of stage 8 and older oocytes. This claim is not supported by their observations since the increase in stage 14 oocytes could result from a combination of continued development of stage 7/8 and older oocytes AND apoptotic elimination of some stage 7/8 and earlier oocytes. I suggest the authors consider this possibility and change their language with respect to the exact stage of oocyst development or do a simple DAPI/caspase staining at different times after infection to quantitate all stages of egg chambers and report amount of early and mid-oogenesis apoptotic events. I understand that this analysis may be outside the scope of the present study and not necessarily critical. The authors may wish not to do this if they change the language so as to consider the possibility that younger oocytes may be eliminated (unless there is some other compelling argument that excludes this possibility).

---

## [Author Response]

*Essential revisions:*

*In general, it is important to strengthen the genetic experiments concerning neuronal manipulations in order to clarify the neuronal contribution to the described phenotypes.*

*1) There is some confusion re: the neuronal site of NF-ΚB activation. Based on the CNS neurites in the images provided, the pLB1-expressing cells seem to include peripheral sensory (taste?) neurons and sensory neurons in the oviduct, but not efferent oviduct-innervating neurons such as the octopaminergic neurons, with cell bodies located in the posterior tip of the VNC (see, for example, Monastirioti et al. (2003) Dev Biol PMID 14623230). Puzzlingly, the authors seem to imply that the octopaminergic neurons are a subset of pLB1 neurons – in fact, the subset that matters for the PGN effects on egg laying. To make matters more complicated, some of the drivers used in the study to demonstrate neuronal effects (e.g. elav-Gal4) also target enteroendocrine cells, recently shown to sense luminal bacteria (see Du et al. (2016) PLoS Genet doi: 10.1371/journal.pgen.1005773). While the authors are careful not to ascribe the observed effects to a specific neuronal subset, the paper will be greatly improved by experiments to restrict and better define the specific octopinergic responsible for the phenotype. The authors should use restricted genetic manipulations and detailed neuronal characterisation. For example, using drivers specific for oviduct sensing (ppk) or efferent (Ilp7, oct+) neurons should allow them to establish the whether the IMD pathway acts in sensory, efferent and/or other endocrine cells to mediate the egg laying reduction.*

As requested by the reviewers, we have tried to more precisely define the neurons implicated in egg laying behavior and responsive to bacteria.

We first try to exclude the possibility that the effect of bacteria on egg laying were due to NF-κB activation in enteroendocrine cells (EE) and not in neurons. Indeed, EE cells have been reported to express one TrpA1(A) transcript that is critical for uracil-dependent defecation and TrpA1 has also been shown to mediate the response to bacteria LPS. To test this hypothesis, we have inactivated the NF-κB pathway by overexpressing UAS Fadd- RNAi specifically in EE cells using the voila- or prospero- Gal4 drivers. The new data presented Figure 5—figure supplement 1 demonstrate that, in contrast to what is happening with the pLB1-Gal4 drivers, NF-κB pathway inactivation in EE cells does not prevent egg laying drop post-infection.

We then, as asked by the reviewers, tested the putative implication of the ppk neurons. Here again, inactivation of the NF-κB in ppk positive neurons did not prevent egg laying drop post infection (these data are now included in Figure 5—figure supplement 1). We also tried to rescue the complete egg laying defect observed in a SPR mutant flies by overexpressing the SPR cDNA in pLB1 cells. As previously published, SPR overexpression under the ppk Gal4 driver was able to fully rescue the egg laying defect of SPR deficient females. This was not the case when we used the pLB1 Gal4 driver demonstrating that pLB1 and ppk cells are genetically different (these data are now presented in Figure 5—figure supplement 1).

Finally, we tested whether the ILP7 positive neurons that innervate the reproductive tract were implicated in the NF-κB dependent response to bacteria. As shown in the new Figure 5—figure supplement 1, contrary to pLB1 cells, NF-κB inactivation in these neurons have no effect on egg laying response to infection.

*2) Related to this point, the authors need to distinguish between a generic role for neurons in controlling egg laying, relaying the PGN-triggered signal or sensing it. In this regard, I see how presenting egg laying phenotypes as their ratio relative to control conditions makes comparisons visually straightforward, but can also mask "constitutive" phenotypes resulting from some genetic manipulations – especially those resulting from interfering with the activity of oviduct-innervating neurons. The authors should clarify which of their genetic manipulations have a strong effect on the constitutive egg laying ability of flies, and which are specific to septic injury responses.*

We agree with the reviewers that the absolute number of eggs laid by females is an important parameter. We choose to show ratios rather than absolute numbers for the following reason. Although flies from a given genotypes are not always laying the same number of eggs over several independent experiments when not infected, the egg lay drop after infection (percentage) is consistent from one experiment to another. We have provided this data to the reviewers as a table showing the absolute number of eggs laid for two independent experiments with several genotypes prior to infection and following bacterial treatment. (See below).

GenotypeMean of the numbers of eggs lay by 3 non infected females in 6 hours (the two numbers correspond to two independent experiments) (for each experiments a minimum of 5 vials containing 3 females is used)Mean of the numbers of eggs lay by 3 infected females in 6 hours (the two numbers correspond to two independent experiments) (for each experiments a minimum of 5 vials containing 3 females is used)Wild type42; 2725; 14Dredd (D55)30; 1728; 16Rel (E20)26; 2527; 25pgrp-LE(112)20; 2817; 22pgrp-LB33; 2920; 15pgrp-LB, pgrp-LC (E12)30; 2110; 4pLB1_Gal4/+40; 3720; 17pgrp-LB-; pLB3_Gal4>Uas_PGRP-LB-PD45; 4122; 17pgrp-LB-; pLB1>Uas_PGRP-LB-PC28; 3817; 21pgrp-LB-; pLB2>Uas_PGRP-LB-PA32; 2514; 14pgrp-LB-; pLB2>Uas_PGRP-LB-PD30; 1916; 9pgrp-LB-; pLB3>Uas_PGRP-LB-PC41; 2725; 15pgrp-LB-; pLB3>Uas_PGRP-LB-PA53; 2040; 12pgrp-LB-; pLB1>Uas_PGRP-LB-PD40; 3330; 31pgrp-LB-; pLB2>Uas_PGRP-LB-PC21; 399; 24Elav_Gal4/+42; 3922; 8Elav_Gal4>Uas_Rel IR45; 4343; 42Elav_Gal4>Uas_Fadd IR32; 2631; 20pLB1_Gal4>uas_Rel IR36; 3430; 34pLB1_Gal4>Uas_LE IR29; 2930; 24Uas_FaddIR/+54; 5237; 33pgrp-LB-PD-(Z10e)25; 1814; 11pgrp-LB-PD-(Z19b)19; 1311; 7TbH_Gal4/+103; 6071; 38Tdc2_Gal4/+100; 5173; 30TbH_Gal4>Uas_Fadd IR78; 2475; 20Tdc2_Gal4>Uas_Fadd IR93; 60100; 63

*3) Based on the data provided, it also seems conceivable that the main effect of PGN is to transiently reduce food intake, which can indirectly impact egg production. Do any of the described genetic manipulations affect one phenotype without affecting the other one?*

In order to respond to this concern, we have measured the food intake of wild type and NF-κB/Relish mutant 6 hours after bacterial infection by pricking. The results obtained, which are now included as Figure 1—figure supplement 3, show that bacterial infection has no significative effect on female feeding behavior. We therefore do not think that the reduction in egg laying is a consequence of a massive reduction in feeding of infected females. This has been added in the text.

*4) The images of ovaries in Figure 1—figure supplement 3 would be more useful if they were stained with a DNA dye that better revealed the mostly translucent very early stage egg chambers, i.e. it's otherwise very difficult to see stage 7/8 and younger egg chamber. Nevertheless, from these few images it is clear that the ovaries 6h post clean injury have vast numbers of very early (stages 2-5/6) whereas 6h post septic injury these early stages are almost completely absent. Because the figure only presents one ovary and the quantitation does not measure earlier stages it is impossible to know what the fate of these earlier stage egg chambers is in response to infection. This is an important point because the authors suggest (Discussion, end of first paragraph) that infection does not trigger oogenesis checkpoints that utilize apoptosis to eliminate early stage oocytes while allowing continued development of stage 8 and older oocytes. This claim is not supported by their observations since the increase in stage 14 oocytes could result from a combination of continued development of stage 7/8 and older oocytes AND apoptotic elimination of some stage 7/8 and earlier oocytes. I suggest the authors consider this possibility and change their language with respect to the exact stage of oocyst development or do a simple DAPI/caspase staining at different times after infection to quantitate all stages of egg chambers and report amount of early and mid-oogenesis apoptotic events. I understand that this analysis may be outside the scope of the present study and not necessarily critical. The authors may wish not to do this if they change the language so as to consider the possibility that younger oocytes may be eliminated (unless there is some other compelling argument that excludes this possibility).*

To better characterize the impact of bacteria infection on oocytes we have performed new quantifications following clean injury and septic injury (6h) using DAPI to better identify specific stages including early ones. By focusing on stage 6, stages 7-9, stage 10 (post-checkpoint) and stage 14 (mature oocytes), we demonstrated that while the very early stages are not impacted by the treatment, there is an accumulation of mature oocytes. In addition, following septic injury, we quantified a significant drop in the amount of stage 10 oocytes (Figure 1—figure supplement 3). We then tested whether this decrease of stage 10 oocytes might be due to apoptosis. For that, we performed anti cleaved-DCP1 antibody staining on ovaries from non-infected, injured only and bacterially infected females. Our data show that the amount of apoptotic events are not increased following septic injury (Figure 1—figure supplement 3) Thus, bacterial infection transiently blocks ovulation, and especially the release of mature oocytes within the oviducts, without inducing a massive apoptosis in ovaries in contrast to other threat such as wasp.